# Isolation of Reporter Cells That Respond to Vitamin A and/or D Using a *piggyBac* Transposon Promoter-Trapping Vector System

**DOI:** 10.3390/ijms23169366

**Published:** 2022-08-19

**Authors:** Kosuke Ishikawa, Sakura Tamamura, Nobuhito Takahashi, Motoki Takagi, Kentaro Semba, Shinya Watanabe

**Affiliations:** 1Japan Biological Informatics Consortium (JBiC), 2-45 Aomi, Koto-ku, Tokyo 135-8073, Japan; 2Medical-Industrial Translational Research Center, Fukushima Medical University, 1 Hikarigaoka, Fukushima 960-1295, Japan; 3Department of Life Science and Medical Bioscience, School of Advanced Science and Engineering, Waseda University, 2-2 Wakamatsu-cho, Shinjuku-ku, Tokyo 162-8480, Japan

**Keywords:** trap vector, reporter cell, gene mining, transposon, vitamin A, vitamin D

## Abstract

Previously, we established a highly sensitive promoter-trapping vector system using the *piggyBac* transposon for the efficient isolation of reporter cells. Herein, we examine whether this screening system can be applied to obtain vitamin-responsive cells. As a result, one and two reporter cells that responded to bexarotene (vitamin A) and calcitriol (vitamin D), respectively, were isolated from 4.7 × 10^6^ seeded HeLaS3 cells. 5′ RACE analyses identified the well-known *CYP24A1* gene as a calcitriol-responsive gene, as well as two new bexarotene- or calcitriol-responsive genes, *BDKRB2* and *TSKU*, respectively. *TSKU*, interestingly, also responded to bexarotene. Endogenous levels of the *TSKU* and *BDKRB2* transcripts displayed only slight changes and were not detected in the comprehensive analyses performed to date. Dose–response analyses of *BDKRB2* and *TSKU* reporter cells in parallel revealed a differential profile in response to each vitamin A agonist, suggesting a bioanalyzer. The present study demonstrates that producing multiple reporter cells by a type of random screening can efficiently identify novel genes with unusual characteristics and be used for the profiling of the properties of vitamin compounds. Similar approaches to the method shown here may be useful for identifying new markers and for the analysis or diagnosis of nutrients, toxins, metabolites, etc.

## 1. Introduction

We previously developed a trap vector system for the efficient isolation of cells that can express reporter genes in response to a certain condition of interest [1] based on transposon-mediated genomic manipulation [2]. In general transposon-mediated vector systems, the transposase gene located between the terminal repeat sequences in natural transposons is replaced with DNA modules, which then can be integrated into the host genome via a cut-and-paste mechanism by a transposase that is expressed via its co-introduction as a helper vector. In our promoter trap system, the DNA modules were designed to express GAL4FF (an extremely trimmed minimal DNA-binding domain of the yeast GAL4 transcription factor with two repeats of the minimal transcription activation module from VP16 [3]), only after they are inserted into the genome, where transcription occurs. Because GAL4FF can lead to the expression of the reporter genes *EGFP* and firefly luciferase (*Fluc*), which lie downstream of the UAS at another site in the vector, even a small amount of promoter activity is amplified and can be detected with high sensitivity [1,4]. Among the transposon systems that are applicable to mammalian cells, we employed the *piggyBac* transposon system because its transposase mutant, called hyPBase, has extremely efficient transposition activity [5]. This *piggyBac* transposase recognizes a short sequence, 5′–TTAA–3′, in the genome during transposition. The 5′–TTAA–3′ sequences are innumerable in the genome, and it is known that the selectivity of the insertion position is random, even though some bias toward transcriptionally active regions has been recognized [6,7]. Therefore, it is possible to deeply explore the trapping of a gene promoter on the genome via a simple vector-introduction operation.

Our trap vector system efficiently produces cells expressing EGFP and Fluc as reporters in response to arbitrary conditions. We reported the production of reporter cells that respond to the expressions of the *c-Myc* gene [1], ER stresses [1], or glucocorticoids [4]. It is not always necessary to identify the responding gene in the reporter cell; however, if so, an unexpected gene is occasionally found as a novel marker. For example, *OSBPL9* was identified as a novel ER-stress–responsive gene, but this gene was overlooked in public data because it ranks fairly low in score [1]. In this way, the almost random screening is highly attractive in identifying marker genes without being bound by past data and models, and without narrowing down multiple candidates. Because of the high signal-to-noise ratio afforded by the amplification mechanism of the GAL4-UAS system, slight changes in responsive gene expressions are extremely highlighted. Therefore, it is possible to select genes as important markers, despite the fact that they show only slight changes in the endogenous levels, which could not be noticed in comprehensive analyses, such as a microarray analysis.

In this study, to examine whether this vector system is also applicable in the field of nutrition, we tried to obtain reporter cells that respond to vitamin A or D, and identified novel responsive genes with low levels of endogenous transcript changes. Among them were genes that responded specifically to each vitamin; interestingly, we also identified a gene that responded to both vitamins. The procedure and results reported here may be an important prototype for other nutrition factors or conditions in the future.

## 2. Results

### 2.1. Cloning of Calcitriol- or Bexarotene-Responsive Cells Using a Transposon-Mediated Promoter-Trapping Vector System

HeLaS3 cells were co-transfected with promoter/enhancer trapping vectors [4] and the transposase (hyPBase, [5]) helper vector. To obtain cells that respond only when reagents are added, cells in which EGFP was constitutively expressed were removed by three rounds of fluorescence-activated cell sorting. Bexarotene or calcitriol was then added, and the cells that expressed EGFP at high levels were single-cell sorted. Finally, three independent clones with a low background level and showing a strong shift of the EGFP reporter signal, as revealed by flow cytometry analysis, were selected (Figure 1).

### 2.2. Identification of Responsive Genes by 5′ Rapid Amplification of cDNA Ends (5′ RACE)

To identify the genes that responded to each vitamin, the transcript of the upstream exon sequence that was fused to *GAL4FF* was analyzed by 5′ RACE (Appendix A), which led to the identification of responsive genes (Table 1). Among them, cytochrome P450 family 24 subfamily A member 1 (*CYP24A1*), which was identified in clone L5#11, is a mitochondrial monooxygenase that initiates the degradation of 1,25-dihydroxyvitamin D3, i.e., calcitriol (the physiologically active form of vitamin D3), by hydroxylation of the side chain, regulates the level of vitamin D3, and is one of the best-known vitamin D-responsive marker genes [8,9]. Conversely, the bradykinin receptor B2 (*BDKRB2*) (clone L20#4) and *TSKU* (*TSK* or *TSUKUSHI*) (clone L3#5) were identified as novel responsive genes to each vitamin. *BDKRB2* encodes a receptor for the 9-aa peptide bradykinin, one of the physiologically active substances that exerts a blood pressure-lowering effect, which is released upon activation by pathophysiological conditions, such as trauma and inflammation [10,11]. The B2 receptor is associated with G proteins that stimulate a phosphatidylinositol–calcium second messenger system [12]. *TSKU* encodes a small leucine-rich proteoglycan and has been predicted or partially demonstrated to be involved in animal organ development, cholesterol homeostasis, bone formation [13], and glycolipid metabolism [14]; however, little is known about its exact functions.

### 2.3. Low Induction Levels of the Newly Identified Responsive Genes Revealed by Real-Time PCR Analysis

To analyze the endogenous expressions of the newly identified genes, *BDKRB2* and *TSKU*, parent HelaS3 cells, in which the trap vector was not introduced, were stimulated using bexarotene or calcitriol, and their mRNA was analyzed by real-time PCR (Figure 2). Consistent with the responsiveness of reporter cells, their genes responded to each vitamin used for screening. However, the fold change in both genes was very low, whereas *CYP24A1* is known to have fold changes that are thousands of times stronger. Interestingly, *TSKU* was found to also be induced by bexarotene, as well as calcitriol (Figure 2, left).

It is known that vitamin A- and D-responsive genes are regulated by nuclear receptor hetero- or homocomplexes, i.e., RAR/RXR and VDR/RXR, respectively, via their recognition of DNA elements, which are termed *RARE* and *VDRE*, respectively. We then examined whether the promoter regions of *BDKRB2* and *TSKU* carry these response elements. According to previous reports [15,16], we searched 5′[GA]G[GT]TCA.{0,8}[GA]G[GT]TCA3′ as *RARE*, where [AB] indicates “A” or “B” nucleotide, “.” indicates any nucleotide, and “.{0,8}” indicates 0–8 of any nucleotide. Similarly, by referring to Ramagopalan et al., we searched 5′[AG][AG]G[GT][TC][CG]A..[GAC][AG]G[TG][TGC][CT][AGT]3′, which is a top-scoring *VDRE* for calcitriol [17]. It was revealed that *BDKRB2* has one *RARE* composed of the two direct repeats of the core hexanucleotide motif (PuG(G/T)TCA), spaced by four nucleotides (DR4) (Figure 3, *BDKRB2_RARE*). Conversely, *TSKU* had not only one *RARE* with no spacer (DR0) (Figure 3, *TSKU_RARE1*), but also three hexanucleotide repeats, each spaced by four nucleotides (Figure 3, *TSKU_RARE2+3*). To the best of our knowledge, genes with this pattern, i.e., two fused *RARE*s (or three PuG(G/T)TCA core repeats), are rare. Moreover, in this case, the rare 4-nt spacer observed between the cores was repeated. In addition to *RARE*, *TSKU* also had three positions that matched the consensus sequence of *VDRE* (Figure 3, *TSKU_VDRE1~3*). Regardless of these numbers and rarities, the existence of the above response sequences was consistent with the reactivity to bexarotene and calcitriol (Figure 2).

### 2.4. Luciferase Assay of Reporter Cells for Responsiveness to Various Vitamin A and D Analogs

To investigate the responsiveness of reporter cells to analogs of vitamins A or D, including drugs approved by the Food and Drug Administration (FDA), we performed a dose–response analysis of these cells using a luciferase assay (Figure 4).

The responsiveness of the L20#4 clone, in which *BDKRB2* was identified, was specific to drugs in the vitamin A group, whereas that of the L5#11 clone (*CYP24A1*) was specific to vitamin D compounds. Conversely, the responsiveness of the L3#5 clone (*TSKU*) was highly interesting, as it responded to both vitamins A and D, as expected. Moreover, the L3#5 clone (*TSKU*) showed a great number of variations in the shape of the dose-dependency curve obtained using vitamin A compounds compared with L20#4 (*BDKRB2*) (Figure 4, left half panels, bold lines). By comparing the fold changes, the order of the vitamin A compounds that reacted strongly differed between clones L3#5 (*TSKU*) and L20#4 (*BDKRB2*) (Figure 5, upper panels). It was also observed that the various vitamin A reagents that exhibited a fold change of approximately 100 times at 1 μM in the L20#4 clone hardly reacted in the L3#5 clone under the same conditions. These results clarify that the drug can be profiled in greater detail than before by using two or more vitamin A-responsive cell lines.

Because *TSKU* showed both vitamin A and D responses, its responsiveness when both vitamins coexisted was finally quantitatively evaluated. It is known that both A and D vitamins bind to RXR; thus, the responsiveness to one vitamin is competitively inhibited by the other vitamin in the case of genes having only a single *RARE* or *VDRE* [18]. Accordingly, in both the luciferase assay for reporter clones (Figure 6a) and the real-time PCR for endogenous mRNA expression (Figure 6b), *BDKRB2* and *CYP24A1* exhibited an inhibitory effect on the response expression when a nonresponsive vitamin coexisted at a high concentration (Figure 6, right panels). In contrast, it was observed that both vitamins additively regulated the responsive expression of *TSKU* (Figure 6, left panels).

## 3. Discussion

In this study, we created reporter cells stimulated by vitamins to test whether the previously developed, highly sensitive trap vector system can be applied in the field of nutrition. This was a challenge because innumerable vitamin-responsive genes have been reported to date by transcriptome analysis, chromatic immunoprecipitation analysis using relevant nuclear receptors, such as VDR, RAR, RXR etc., and in silico analyses which explore response sequences [15,19,20,21]. Nevertheless, among the three causative genes identified in the isolated reporter cells by 5′ RACE, two (*BDKRB2* and *TSKU*) were novel genes regulated by vitamins (Table 1). These genes both had a DR4-type *RARE*, which is a rare pattern in the literal sense, in their promoters (Figure 3). Thus, it would probably have been difficult to perform in silico sequence searches based on conventional wisdom (for example, the so-called 3-4-5 rule (refs in [22]) might have been an obstacle). Additionally, the changes in their endogenous mRNA levels were at most only about 2–3-fold, even with the most inducible concentration of the reagent (Figure 2, compared to Figure 4), whereas that of the well-known *CYP24A1* gene was changed by several thousand-fold (Figure 6b). If we tried to narrow them down from comprehensive analyses, such as a microarray analysis, using such a low fold change, it would be absolutely impossible to classify them (*BDKRB2* and *TSKU*) as markers. Therefore, we consider that the method of almost randomly inserting a reporter gene into the intrinsic gene region and letting it discover responsive genes is a more efficient and attractive method for identifying new marker genes in any condition of interest.

The observation that the reactivity to vitamin A and its analogs varied between L3#5 (*TSKU*) and L20#4 (*BDKRB2*) (Figure 4, left panels and Figure 5, upper panels) led to several interpretations. One was that the number of *RARE*, their positions, and *cis* elements, as well as the variation in the distance of the spacer between the cores in the *RARE* in each promoter, may confer different “recognition eyes” to substances. Models analogous to this have been reported in which individual genes differ in their responsiveness to vitamin D due to differences in multiple VDREs and chromatin structure [23]. Another was that the responding cells rigorously recognized the “difference” among them, not only as vitamin A. It is unclear at this stage how this difference was recognized, but the following scenarios can be imagined. First, the complex of each vitamin A with nuclear receptors and *RARE* forms a slightly different structure, which may change transcriptional activity or efficiency. Second, it is possible that only minor structural differences act on different targets other than vitamin receptors, and feedback from them may exert effects on promoters/enhancers or epigenetics. Nevertheless, we established that, using the two cell lines, a profile characteristic of each compound was obtained as a pattern of dose–response curves. This is reminiscent of the unique spectrum of each of the fluorescent proteins. By increasing the repertoire of these reporter cell lines, it may be possible to create something like a bioanalyzer system. In this study, we obtained reporter cells limited to vitamins; however, this system could also be applied to nutrients, toxins, metabolites, etc.

## 4. Materials and Methods

### 4.1. Establishment of Reporter Cells Responsive to Bexarotene or Calcitriol

HeLa cells were plated into 6-well or 12-well plates at a concentration of 1.3 × 10^5^/mL. On the following day, the cells were incubated with a mixture of 1 μg of DNA of the hyPBase, and the transposon donor vector (at a ratio of 1:3) and 4 μg of polyethylenimine (Polysciences, Warrington, PA, USA) in 100 μL of Opti-MEM for 1 mL of the culture volume. After cell propagation, the EGFP-positive population, which was considered to be expressed constitutively, was removed by three rounds of cell sorting on an SH800Z cell sorter (Sony, Tokyo, Japan). Next, the cells propagated on a 10-cm dish scale were stimulated with 10 μM bexarotene (No. 11571, Cayman Chemical, Ann Arbor, MI, USA) or 150 nM calcitriol (D1530, Sigma, Tokyo, Japan) overnight at 37 °C in a CO_2_ incubator; subsequently, those that appeared in the EGFP-high fraction were single-cell sorted into 96-well plates. After propagation of those cells, their reagent responsiveness was checked under a fluorescence microscope. Whenever multiple clones were obtained from the same parental cells, the genomes were subjected to a splinkerette PCR analysis, as described in [1], to select independent clones as assumed from their band patterns.

### 4.2. Identification of Responsive Genes by 5′ Rapid Amplification of cDNA Ends

5′ RACE was performed exactly as described previously [4].

### 4.3. Real-Time PCR Analysis

RNA was extracted from the cultured cells using the ISOGEN reagent (Nippon Gene, Tokyo, Japan). Reverse transcription was performed using the SuperScript III First-Strand Synthesis Kit (Thermo Fisher Scientific, Waltham, MA, USA) with random hexamers as the primer mix. The THUNDERBIRD SYBR qPCR Mix (TOYOBO, Osaka, Japan) was used for real-time PCR, which was executed on a StepOnePlus system (Applied Biosystems, Waltham, MA, USA). The primers used in this study are listed in Table 2. The human *HPRT1* gene was used as the internal control.

### 4.4. Luciferase Assay

The luciferase assay was performed using a slightly modified version of the protocol originally described in the study by Siebring-van Olst [24]. Briefly, cells were spread on a white-colored 96-well cell culture plate. Eighteen hours after the addition of the stimulating reagent (for a total volume of 90 μL), the plate was set on a plate reader (Berthold Technologies, TriStar^2^S LB942) and the following program was executed: 90 μL of 2 × FLAR (40 mM Tricine [pH8], 200 μM EDTA, 2.2 mM MgCO_3_, 5.3 mM MgSO_4_∙7H_2_O, 500 μM ATP, 2% TritonX-100, 40 mM DTT, and 500 μM d-luciferin) was dispensed to all wells of the plate, which was then shaken for 180 s, followed by photon counting for 3 s per well after a 0.2 s delay. The means of three independent experiments (in each independent experiment, three samples were measured for one condition (*n* = 9)) were graphed using GraphPad Prism software. Vitamin drugs (tretinoin (ATRA) (S1653), isotretinoin (S1379), acitretin (S1368), adapalene (S1276), tamibarotene (S4260), tazarotene (S1569), alfacalcidol (S1468), and doxercalciferol (S1467)) were purchased from Selleck. Calcitriol (1α,25-dihydroxyvitamin D3) (D1530) was purchased from Sigma. Bexarotene (No. 11571) and 9-cis retinoic acid (9-cis RA) (No. 14587) were purchased from Cayman Chemical. In the experiment reported in Figure 6, which aimed to investigate the responsiveness in the presence of both vitamin A and vitamin D, the addition of the reagents was performed simultaneously.

## 5. Conclusions

Our previously developed, highly sensitive promoter trap system was shown to facilitate the production of vitamin-responsive reporter cells, which led to the identification of two new vitamin-A-responsive genes, *TSKU* and *BDKRB2*. Both of these endogenous genes exhibited only slight changes in response to vitamin A and were not recognized as markers in the past. By analyzing the obtained multiple reporter cells in parallel, various agonists could be classified based on the dose–response curves. Because this screening system is highly versatile and easy to execute and has a high potential to be applied to other physiologically active substances and drugs, the results reported here seem to be of importance and may be a prototype for future research in various fields.

## Figures and Tables

**Figure 1 ijms-23-09366-f001:**
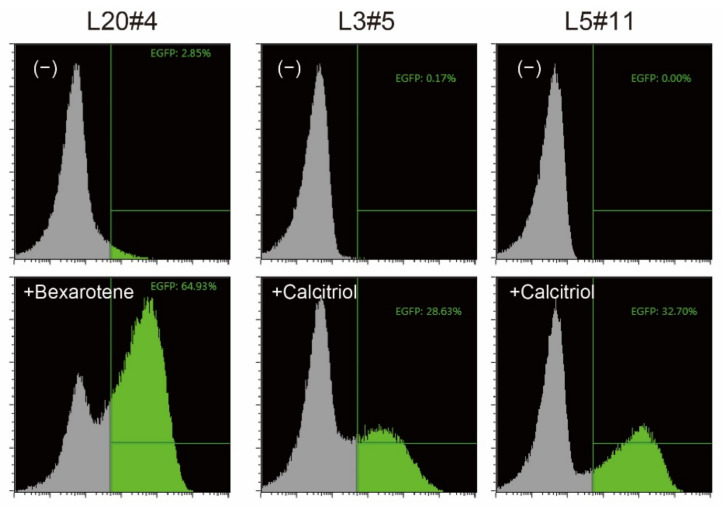
Confirmation of reporter cells. The isolated clones were treated without (−) or with the indicated vitamin compound for 18 h, and the expression of EGFP, one of the reporter proteins, was analyzed by flow cytometry. EGFP-positive cells are shown in green.

**Figure 2 ijms-23-09366-f002:**
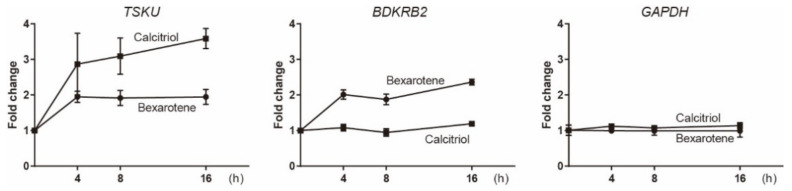
Real-time PCR analysis of the expressions of endogenous *TSKU* and *BDKRB2* upon stimulation by 10 μM bexarotene or 1 μM calcitriol. *HPRT1* was used as the internal control. The graphs show the mean of two experiments per gene with three measurements each. Error bars indicate SD.

**Figure 3 ijms-23-09366-f003:**
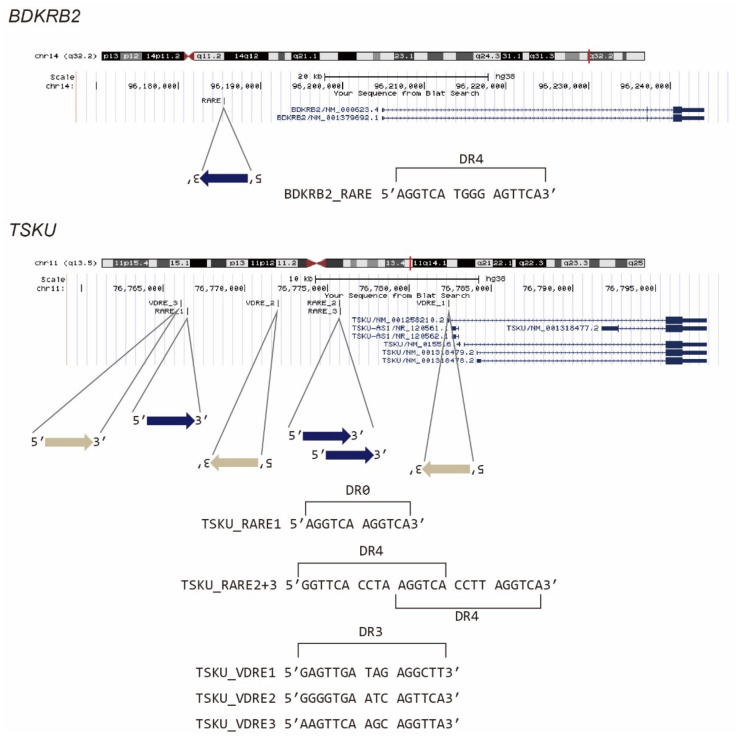
Presumed vitamin-responsive genomic sequences of *BDKRB2* and *TSKU*. The known consensus sequences of *RARE* or *VDRE* were searched in a promoter area of about 20 kb, and the position of the hit sequence was mapped using the UCSC genome browser.

**Figure 4 ijms-23-09366-f004:**
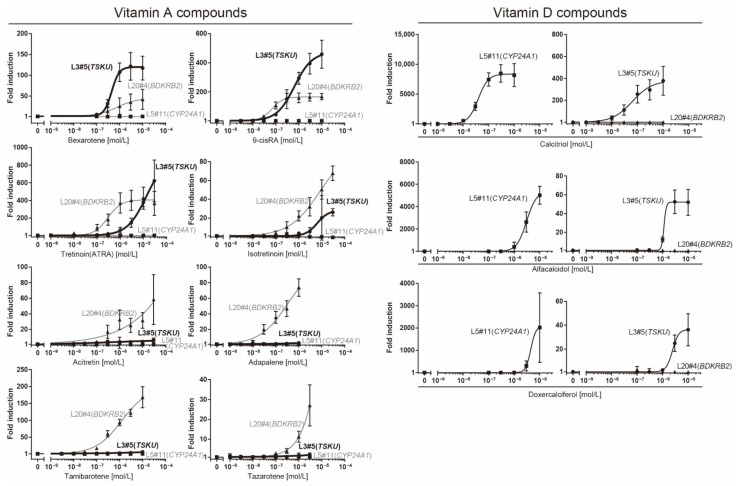
Dose–response analysis of reporter cells (Table 1) against various vitamin analogs using a luciferase assay. The plots are the mean from three independent experiments with three measurements each. Error bars indicate SD. The fold-induction value for 0 M was defined as 1.

**Figure 5 ijms-23-09366-f005:**
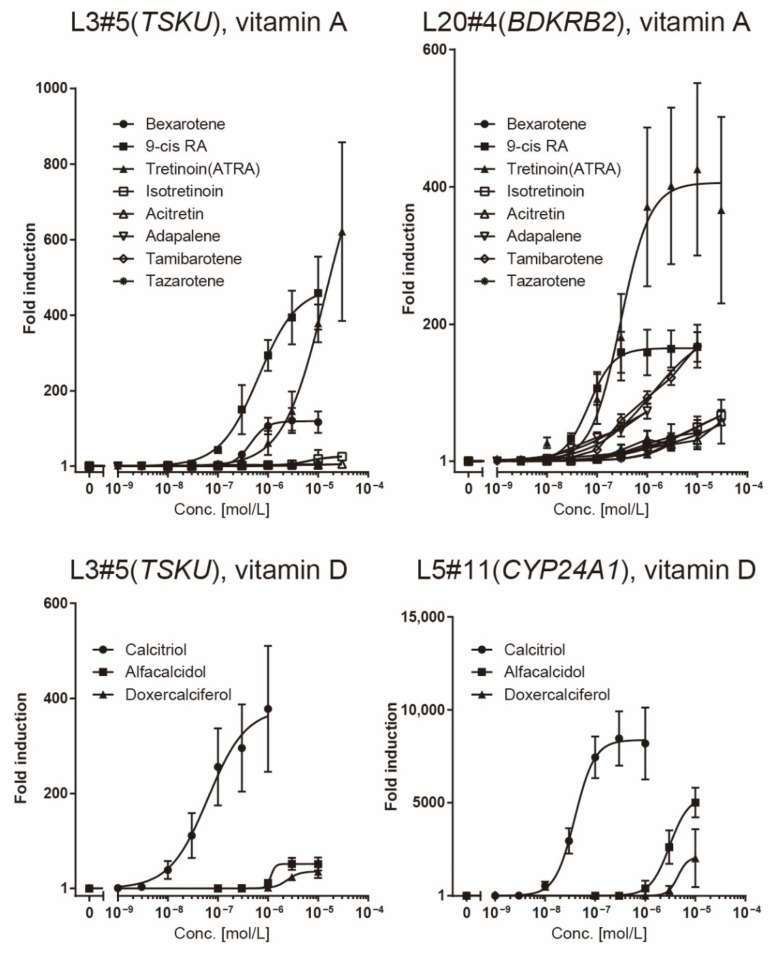
Comparison of the responsiveness of reporter cells to vitamin reagents. The data presented in Figure 4 was redrawn for comparison of responsiveness between the reagents in each reporter cell.

**Figure 6 ijms-23-09366-f006:**
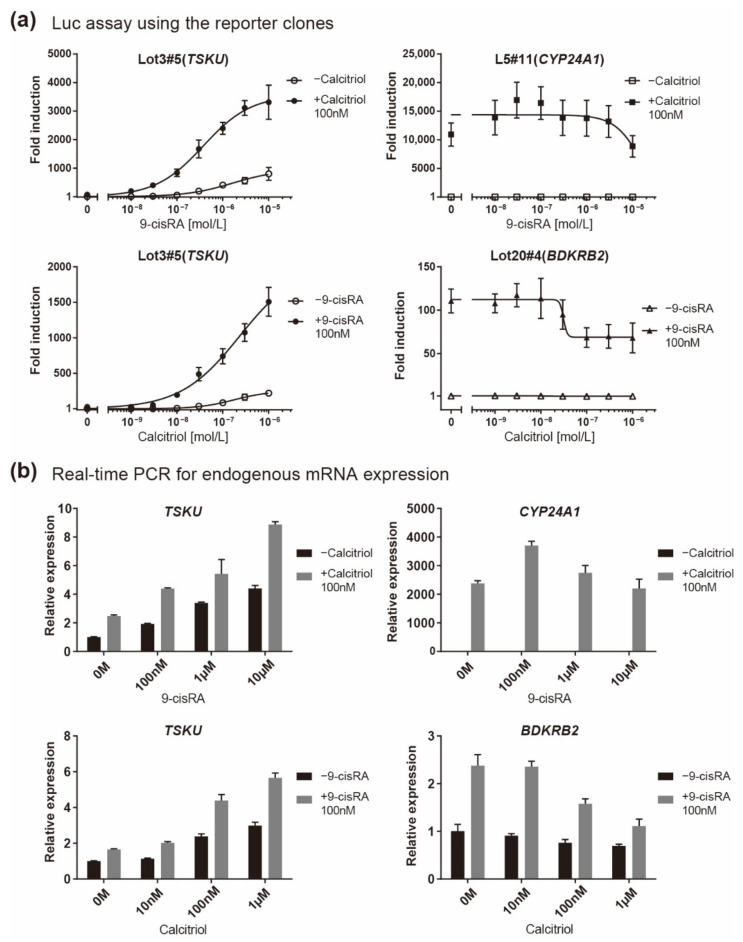
Responsiveness in the case of co-existence of vitamin A (9-cisRA) and vitamin D (calcitriol). A quantitative analysis was performed using two assays when the concentration of 9-cisRA was increased in the absence or in the presence of calcitriol and vice versa. (**a**) Luciferase assay of the indicated reporter cell clone in which the gene indicated in parentheses responded and (**b**) real-time PCR analysis of the endogenous expression of the indicated mRNA. The plots are the mean from three independent experiments with three measurements each. Error bars indicate SD.

**Table 1 ijms-23-09366-t001:** List of bexarotene- or calcitriol-responsive reporter cell clones and responsive genes, as revealed by 5′ RACE.

Clone#	Reagent Used for Screening	Responsive Gene
L20#4	Bexarotene (vitamin A)	*BDKRB2*
L3#5	Calcitriol (vitamin D)	*TSKU*
L5#11	Calcitriol (vitamin D)	*CYP24A1*

**Table 2 ijms-23-09366-t002:** Primers used for the real-time PCR.

Gene	Primer Sequence (5′→3′)
*TSKU*	CTGAGCGACGTGAACCTTAGC
	CCTGACTGTGCGTCGTGAAG
*BDKRB2*	GTACCAGGGAGCGACTGAAG
	GGGCAAAGGTCCCGTTAAGA
*CYP24A1*	CTCATGCTAAATACCCAGGTG
	TCGCTGGCAAAACGCGATGGG
*GAPDH*	GAAGGTGAAGGTCGGAGTC
	GAAGATGGTGATGGGATTTC
*HPRT1*	TGACACTGGCAAAACAATGCA
	GGTCCTTTTCACCAGCAAGCT

## Data Availability

Not applicable.

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
