# Peer review of "Isolation of Reporter Cells That Respond to Vitamin A and/or D Using a piggyBac Transposon Promoter-Trapping Vector System"

_ijms, 2022, doi:10.3390/ijms23169366_

Round 1

Reviewer 1 Report

The manuscript expands upon an earlier paper by the group, using a piggyBac transposon, promoter-trapping approach to generate sensitive reporter cell lines. In that paper they used the approach to generate reporter cell lines responding to glucocorticoids. In the present paper, they used the same approach to generate reporter cell lines responding to vitamin D or A. The paper represents a useful advance in the field and I would recommend publication after a few minor recommendations:

  1. The paper is generally under-referenced. For example, CYP24A1 (lines 93-97) and BDKRB2 (lines 99-103) are introduced and functionally described without reference. Please re-read the manuscript specifically to address the problems with referencing.
  2. qRT-PCR analysis of the endogenous expression levels of TSKU and BDKRB2 were performed at levels of calcitriol (1μM) much higher than used in reporter screening (150nM). The description in Results (lines 115-116) should be expanded to make this point explicit.  
  3. Figure legends need to state the number of biological and/or technical replicates, as well as what is signified by the error bars (+/- SEM, SD?)
  4. Figure 4 would be more easily interpretable if the X-axes were kept constant.
  5. Figure 6A lists the promoter-trapped genes corresponding to the arbitrarily labelled cell lines. This is a benefit to the reader, which would be appreciated also in Figures 4 and 5.
  6. Append the 5’RACE sequences in the Supplementary Information. 

Author Response

Point 1: The paper is generally under-referenced. For example, CYP24A1 (lines 93-97) and BDKRB2 (lines 99-103) are introduced and functionally described without reference. Please re-read the manuscript specifically to address the problems with referencing.

Response 1: Thank you for pointing this out. We have reviewed the entire document and inserted the appropriate references.

Point 2: qRT-PCR analysis of the endogenous expression levels of TSKU and BDKRB2 were performed at levels of calcitriol (1μM) much higher than used in reporter screening (150nM). The description in Results (lines 115-116) should be expanded to make this point explicit.

Response 2: Since the results in Figure 4 show that the responsiveness of reporter cells does not differ between the 150 nM and 1 μM concentration ranges of calcitriol, I believe it is not necessary to specify that they differ. Unfortunately, we were unable to use the concentration 1 μM for reporter screening as a matter of cost.

Point 3: Figure legends need to state the number of biological and/or technical replicates, as well as what is signified by the error bars (+/- SEM, SD?)

Response 3: We had added the missing legends.

Point 4: Figure 4 would be more easily interpretable if the X-axes were kept constant.

Response 4: We have aligned the horizontal axis as advised for easy interpretation.

Point 5: Figure 6A lists the promoter-trapped genes corresponding to the arbitrarily labelled cell lines. This is a benefit to the reader, which would be appreciated also in Figures 4 and 5.

Response 5: We have changed the labels of Figures 4 and 5 to the easier-to-read notation as in Figure 6A (a).

Point 6: Append the 5’RACE sequences in the Supplementary Information.

Response 6: We have added the sequence data described in Figure S1 and S2 to the supplement file (Supplementary Data S1–5).

Reviewer 2 Report

The manuscript entitled „Isolation of reporter cells that respond to vitamin A and/or D using a piggyBac transposon promoter-trapping vector system” presenst screening system using the piggyBac transposon to obtain vitamin-responsive cells.

In my opinion the manuscript is well written, interesting, and may be very helpful in vitamin research.

My comments/questions:

1.      Why cells were treated for 18 h?

2.      Discussion section should be discussed with other researches – this section should be improved.

Author Response

Point 1: Why cells were treated for 18 h?

Response 1: Thank you for your comments and valuable suggestions. This time frame was considered appropriate as it allowed for an experimental schedule with low error and was a time span for which concentration-dependent curves could be drawn.

Point 2: Discussion section should be discussed with other researches–this section should be improved.

Response 2: A description of other studies that are highly relevant has been included in the Discussion section in the revised manuscript.